# MoRA: Mobility as the Backbone for Geospatial Representation Learning at Scale

**Ya Wen[1]\*, Jixuan Cai[2]\*, Qiyao Ma[1], Linyan Li[3], Xinhuan Chen[2], Chris Webster[1], Yulun Zhou[1]†**

[1]The University of Hong Kong    [2]WeChat, Tencent    [3]City University of Hong Kong

`wenya@connect.hku.hk, codyjcai@gmail.com, qyma@connect.hku.hk,`
`linyanli@cityu.edu.hk, chenxinhuancxh@gmail.com,`
`cwebster@hku.hk, yulunzhou@hku.hk`

## Abstract

Representation learning of geospatial locations remains a core challenge in achieving general geospatial intelligence, with increasingly diverging philosophies and techniques. While Earth observation paradigms excel at depicting locations in their physical states, we propose that a location's full characterization requires grounding in both its physical attributes and its internal human activity pattern, the latter being particularly crucial for understanding its human-centric functions. We present MoRA, a human-centric geospatial framework that leverages a human mobility graph as its core backbone to fuse various data modalities, aiming to learn embeddings that represent the socio-economic context and functional role of a location. MoRA achieves this through the integration of spatial tokenization, GNNs, and asymmetric contrastive learning to align 100M+ POIs, massive remote sensing imagery, and structured demographic statistics with a billion-edge mobility graph, ensuring the three auxiliary modalities are interpreted through the lens of fundamental human dynamics. To rigorously evaluate the effectiveness of MoRA, we construct a benchmark dataset composed of 9 downstream prediction tasks across social and economic domains. Experiments show that MoRA, with four input modalities and a compact 128-dimensional representation space, achieves superior predictive performances than state-of-the-art models by an average of 12.9%. Echoing LLM scaling laws, we further demonstrate the scaling behavior in geospatial representation learning. We open-source code and pretrained models at: https://github.com/ylzhouchris/MoRA.

## 1 Introduction

At the heart of Geospatial Artificial Intelligence (GeoAI) lies a fundamental challenge: the translation of a geographic location—a complex, multifaceted entity defined by physical features, human activity, and latent relationships—into a dense, low-dimensional vector that a machine can process and reason with (Chen et al., 2025; Zhang et al., 2024; Mai et al., 2025). Such a vector, or "representation", enables models to understand spatial patterns, predict socio-economic outcomes, and track dynamic changes. However, the path toward creating these representations has split into two dominant paradigms, each attempting to answer different fundamental questions about the nature of a place. This divergence shapes every subsequent decision, from data acquisition and model architecture to the very definition of what a location embedding should represent.

The first major paradigm centers on Earth observation. It seeks to describe the "physical" state of locations and build a consistent digital twin of the planet's surface. Such digital representations provide richer geographic priors than raw coordinates, offering location context that improves downstream tasks such as fine-grained image classification(Mac Aodha et al., 2019; Cole et al., 2023). A standout example within this paradigm is Google's AlphaEarth (Brown et al., 2025), which ingests petabytes of raw sensor data from a multitude of sources, i.e., optical satellites, radar, LiDAR,

---

\*Equal contribution. Work done when Ya Wen was an intern at Tencent.
†Corresponding author.

and regularizes this heterogeneous information into a unified representation. Other works learns embeddings by aligning coordinates with visual imagery that delineates the physical appearance of locations, either geo-tagged photos or satellite imagery, using contrastive learning (Vivanco Cepeda et al., 2023; Mai et al., 2023a; Klemmer et al., 2025).

The human-centric paradigm stands out, in sharp contrast, as the second major stream, built on the premise that a location's meaning is defined more by the intricate patterns of human activity that unfold within it. Existing studies have employed various types of human-related data, such as human mobility data (Zhou et al., 2023; Yong & Zhou, 2024) and demographics (Wen & Zhou, 2024), to model socio-economic dynamics and predict outcomes including housing prices, crime patterns, and consumption levels (Xu & Zhou, 2024; Zhang et al., 2022). However, developing an effective approach, especially grounded by a robust and principled rationale, to utilizing inherently multimodal geospatial data remains a significant and unresolved challenge. There is also a lack of comprehensive benchmarks for human-centric tasks, making it difficult to evaluate the inference capabilities of location representations.

To address these challenges, we claim that the true, latent meaning of a location is primarily defined by its functional relationships with other locations, as revealed through the patterns of human movement, instead of its intrinsic attributes. The theoretical justification is explicitly drawn from an analogy to the functioning of Large Language Models (LLMs), where the semantic meaning of a word (a "token") is not inherent but emerges from its context—the other words with which it frequently co-occurs in vast text corpora. Similarly, inspired by Vision Transformers (ViT), we discretize continuous geographic space into cells, treating these discrete grid cells as "spatial tokens" and human mobility—sequences of movements from one cell to another—as the "sentences" that provide context. A cell's representation is thus profoundly enriched by the latent co-occurrence structures revealed in the sequence-like human movement patterns.

To this end, we propose **MoRA**, a framework that leverages mobility as the backbone for geospatial representation learning at scale. By using a human mobility graph as the central structural backbone for multimodal data alignment, MoRA ensures that all other data modalities are interpreted through the lens of these fundamental human dynamics. Our contributions can be summarized as follows:

- We emphasize mobility's fundamental role as the "syntax of geospatial space", which gives functional meaning to region tokens by structuring them in sequences, and propose **MoRA**, a framework that uses a solid relational backbone (mobility) enriched by functional (POIs), social (demographics), and physical (imagery) attributes to learn uniquely comprehensive, humen-centric location representations.

- Drawing a parallel to the large-scale training paradigms in LLMs, we apply and validate MoRA using a massive mobility graph with extensive long-range links, yielding high-quality, transferable region representations at the national scale. We also provide, to our knowledge, the first empirical evidence of **scaling laws in geospatial representation learning**.

- We construct a **benchmark evaluation dataset** of 9 downstream socio-economic prediction tasks. Extensive experiments show that our model achieves significantly superior performance—improving by 12.9% on average—across all tasks.

- We open-source our methodology, a benchmark dataset for human-centric representation evaluation, and a **distilled version of MoRA** presented as a simple, privacy-preserving, ready-to-use utility (coordinates in, vectors out) for national-scale geospatial inference.

## 2 RELATED WORKS

**Representation of the physical states of locations.** Several studies encode the natural and built environmental features of locations into machine-interpretable representations, enabling analysis of natural phenomena and monitoring of environmental changes (Brown et al., 2025; Klemmer et al., 2025). Such approaches often rely on geo-tagged and satellite images, or more broadly on Earth observation data (Yin et al., 2019; 2021). CSP (Mai et al., 2023a) learns embeddings from unlabeled imagery with a contrastive objective, but is pretrained, fine-tuned, and tested on the same datasets (iNat2018 (Van Horn et al., 2018) and fMoW (Christie et al., 2018)), without demonstrating generalizability to other tasks. SatCLIP (Klemmer et al., 2025) and GeoCLIP (Vivanco Cepeda et al., 2023) both train location encoders via contrastive learning on coordinate-image pairs, differing

mainly in encoder architectures and pretraining data. Despite showing success in specific tasks such as worldwide image geo-localization, such visual models may miss crucial non-visual context.

**Pretrained foundation model for human-centric geospatial inference.**   Geospatial data is inherently diverse and multimodal, spanning maps (Balsebre et al., 2024; Choudhury et al., 2025), texts (Li et al., 2022; 2023; Feng et al., 2025), images (Fan et al., 2023; Huang et al., 2023), and graphs (Wu et al., 2022; Wang & Li, 2017), all widely employed to advance the understanding of urban areas. To leverage this diversity, multimodal region representation learning has emerged as a paradigm that integrates cross-modal information in a self-supervised manner to learn task-agnostic regional embeddings (Yan et al., 2024; Zhang et al., 2020; 2022; Yong & Zhou, 2024). However, such models are often trained separately for each city and show substantial performance drop when transferred across cities—indicative of sensitivity to domain shifts and of gaps in learning city-agnostic geospatial priors. Following the success of pretrained Foundation Models (FMs) in language and vision domains, there is increasing interest in foundation models for geospatial artificial intelligence (GeoAI) (Mai et al., 2024; Zhang et al., 2024). ReFound (Xiao et al., 2024) is an early attempt that distills knowledge from general FMs while adding domain-specific objectives, but its pretraining data, i.e., POIs and satellite image data from five major Chinese cities, are limited in scope and modality diversity compared with ours. Conceptually closest to our work is the Population Dynamics Foundation Model (PDFM) (Agarwal et al., 2024), which encodes postal codes and counties across the United States by leveraging human activity signals such as regional busyness and search trends. Despite advances in scope, it still falls short in modality richness and diversity. More importantly, it fuses modalities through simple concatenation and aggregates information based solely on a static neighborhood graph defined by geographic proximity. This approach misses the dynamic, non-local, and higher-level correlations between regions.

**Position encoding functions.**   A straightforward way to represent geolocations is to analytically map 2D/3D coordinates into higher-dimensional vectors that preserve geographical distances. Space2Vec (Mai et al., 2020) encodes locations using multi-frequency sinusoid functions in a 2D Euclidean space. Neural Radiance Fields (NeRF) (Mildenhall et al., 2021) encodes continuous 3D coordinates via Fourier input mapping. To model Earth's curvature, Sphere2Vec (Mai et al., 2023b) and Siren (Rußwurm et al., 2024) use position encoders that capture spherical distances between coordinates. These learning-free position encoders are typically followed by a neural network, trained in a supervised manner on task-specific data.

## 3   METHODOLOGY

The framework learns versatile region representations by combining spatial tokenization, Graph Neural Networks (GNNs), and asymmetric CLIP-based contrastive learning. It adopts a mobility-as-backbone architecture that aligns diverse geospatial modalities to a mobility anchor: Points of Interest (POIs) as texts, satellite imagery as visual data, and demographic distributions as tabular categorical histograms. Figure 1 illustrates the proposed methodological framework.

### 3.1   MOBILITY AS THE BACKBONE

Geographic location provides a precise anchor for aligning multimodal data. Prior region embedding methods (Wu et al., 2024; Klemmer et al., 2025) often fuse modalities via coordinates, focusing solely on imagery, which limits the incorporation of relational signals. Human activity routinely spans geographic boundaries via transport or digital networks, and mobility data—represented as links in a spatial graph—captures these non-local patterns. We propose a mobility graph as the backbone for multimodal fusion and alignment in geospatial representation learning.

**Tokenizing Locations for Graph.**   Raw spatial interactions are sparse and misaligned in time and space, making coordinate-based graphs too fragmented to reflect major flows. Inspired by Vision Transformers (ViT) (Khan et al., 2022), which partition images into patches, we grid geographic space and build graphs at the cell level. We adopt the H3 grid (Uber Technologies, Inc.), which minimizes distortion compared to Geohash and Google S2 (Jiang & Zhou, 2024), offering balanced global coverage. At level 6 ($\sim$36 km² per cell), we construct a nationwide mobility graph from real-world transaction data as presented in the "Graph Construction" section in Figure 1.

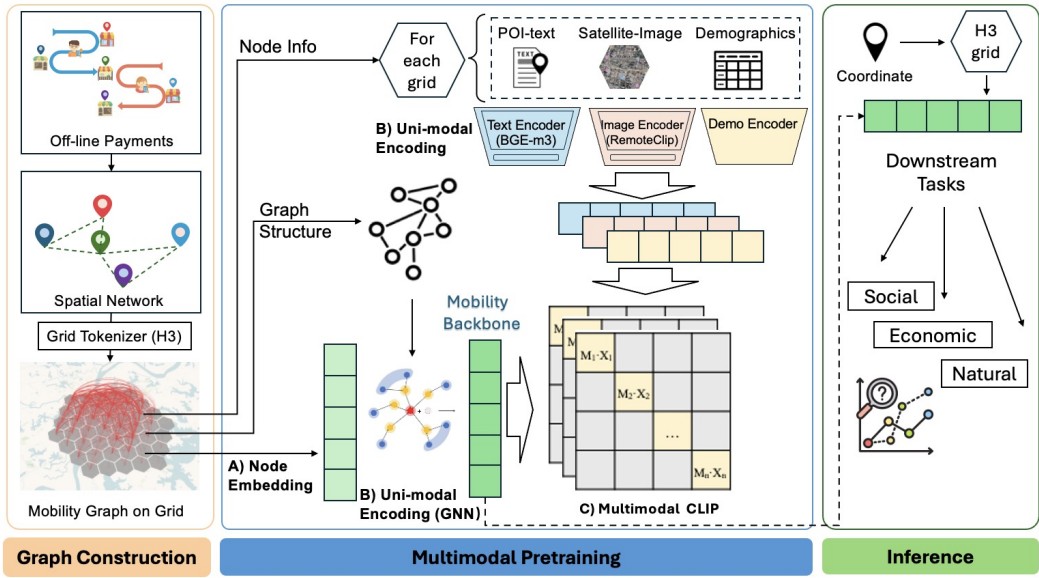

Figure 1: **The methodological framework of MoRA.**

**Pre-encoding.** We utilize LINE (Large-scale Information Network Embedding) (Tang et al., 2015) to preprocess the full mobility graph and generate 128-dimensional embeddings for each H3 cell. We select LINE embeddings with second-order proximity due to their superior performance in empirical evaluation. Paired with the graph sampling strategy described below, this approach effectively preserves full-graph information with minimal computation.

**Graph-based mobility encoder.** Graph Neural Networks (GNNs) have proven to be effective for capturing relationships between nodes, taking into account high-order dependencies. GNNs operate through a message-passing mechanism, where each node updates its representation by aggregating information from its neighbors. To encode a graph $\mathcal{G}$ using $L$ layers of GNNs, the $l^{th}$ layer embedding of a node $i$ is computed as follows:

$$e_i^{(l)} = AGG\left(e_i^{(l-1)}, \{e_j^{(l-1)} \mid j \in \mathcal{N}_i\}\right) \tag{1}$$

where $\mathcal{N}_i$ refers to the neighborhood of the node $i$, and $AGG(\cdot)$ is the aggregation function, which can vary across different models. We use LightGCN (He et al., 2020) as the mobility graph encoder. It simplifies the traditional GNN architecture by removing feature transformation and nonlinear activation, and only keeping the most essential component: neighborhood aggregation. In LightGCN, node features are initialized as learnable parameters, and $L$ layers of propagation yield $L + 1$ embeddings per node, namely $(e^{(0)}, e^{(1)}, \ldots, e^{(L)})$:

$$\mathbf{e}_i^{(l+1)} = \sum_{j \in \mathcal{N}_i} \frac{1}{\sqrt{|\mathcal{N}_i|}} \mathbf{e}_j^{(l)} \tag{2}$$

The final embeddings are computed by summing all layer embeddings:

$$\mathbf{e}_i = \sum_{k=0}^{K} \mathbf{e}_i^{(k)} \tag{3}$$

We initialize node features with pre-encoded LINE embeddings and feed a mobility subgraph into the encoder. This subgraph is extracted from the full graph using a *top-k* sampling strategy based on a percentage threshold, ensuring that the links with the highest flows are retained for each node. This strategy aims to address the "long-tail" distribution patterns commonly seen in geospatial studies, where a small number of regions experience very high flows, while the majority of regions exhibit significantly lower flows that have minimal impact on the overall graph structure. We adopt a top sampling ratio of 10% for the main experiments.

## 3.2 AUXILIARY MULTIMODALITY ENCODER

Different unimodal encoders are selected to encode auxiliary modalities. **(1) Point-of-Interest (POI):** We encode both POI categorical statistics and textual information with the BGE-m3 language model (Chen et al., 2024; Cheng et al., 2025), averaging all POI embeddings within each H3 cell to obtain the grid embedding. **(2) Satellite imagery:** We use an off-the-shelf pretrained RemoteClip (Liu et al., 2024) encoder to produce grid-level image embeddings. **(3) Demographic Statistics:** Categorical demographic tabular data is encoded with an MLP-based encoder.

## 3.3 MULTIMODALITY ALIGNMENT WITH MOBILITY

We train with CLIP-inspired contrastive learning scheme: all four modalities are projected into a shared space, with mobility as the anchor modality to which the others are aligned. We construct modality tuples $(\mathcal{M}, \mathcal{I}, \mathcal{T}, \mathcal{D})$ for all grids in the study area, where $\mathcal{M}$ indicates mobility, and $\mathcal{I}, \mathcal{T}, \mathcal{D}$ represent images, texts, and distribution data, respectively. Given the mobility feature $\mathbf{m}_i$ of a grid and its corresponding observations in other modalities $\mathbf{I}_i, \mathbf{t}_i, \mathbf{d}_i$, we encode them into normalized embeddings by modality-specific encoders: $f(\mathbf{m}_i) \in \mathbb{R}^d$ and $g(\mathbf{x}_i) \in \mathbb{R}^d$ where $\mathbf{x} \in \{\mathbf{I}, \mathbf{t}, \mathbf{d}\}$. Specifically, $f$ is a graph neural network, whereas $g$ are pretrained foundation models for texts and images (see Section 3.2), and an MLP for demographic categorical features. The encoders are optimized with the objective

$$\mathcal{L} = \frac{1}{|\{I, t, d\}|} \sum_{\mathcal{X} \in \{I, t, d\}} (\mathcal{L}_{\mathcal{M}, \mathcal{X}} + \mathcal{L}_{\mathcal{X}, \mathcal{M}}) \tag{4}$$

where the three assisting modalities are aligned with mobility individually and simultaneously by

$$\mathcal{L}_{\mathcal{M}, \mathcal{X}} = \frac{1}{2N} \sum_{i=1}^{N} -\log \frac{\exp(\langle f(\mathbf{m}_i), g(\mathbf{x}_i) \rangle / \tau)}{\sum_{j=1}^{N} \exp(\langle f(\mathbf{m}_i), g(\mathbf{x}_j) \rangle / \tau)} \tag{5}$$

which maximizes the similarity between a grid's mobility features and its corresponding POI, image, or demographic features, while minimizing similarity to those of other grids in the same batch $(\mathbf{m}_i, \mathbf{x}_i)_{i=1}^{N}$, and a symmetric loss $\mathcal{L}_{\mathcal{X}, \mathcal{M}}$ doing the other way around. $\langle \cdot, \cdot \rangle$ represents the dot product operation and $\tau$ is a temperature parameter adjusting the smoothness of the logits in the softmax.

## 4 EXPERIMENTS

### 4.1 EXPERIMENTAL SETTINGS

#### 4.1.1 MULTIMODAL PRETRAINING

**Mobility dataset.** We construct a spatial network using mobility flows derived from Tencent's electronic payment ecosystem, a nationwide transaction platform in China with extensive offline coverage. Each offline store is mapped to an H3 grid cell. By linking cells based on shared weekly transactional traffic across different grids, we form a weekly mobility graph. Aggregating 54 weeks across 50 million stores yields a large-scale graph at H3 resolution level 6, with approximately 200,000 nodes and 1.2 billion edges.

**Auxiliary modality datasets.** We leverage 100M+ Points-of-Interest (POIs) from Tencent Maps, each with a name and category label spanning 445 categories (Tencent LBS Platform, 2024). POI names are typically short, referring to store names, company names, or types of public facilities. For imagery, we fetch Google Satellite imagery at approximately 10m resolution (zoom level 14) for each H3 grid. Demographic statistics are derived from WorldPop (Bondarenko et al., 2025) by aggregating population count per H3 grid, stratified by 10-year age bands, gender, and work-home status in 2020.

**Implementation details.** We split the dataset into 90% for training and 10% for validation to mitigate overfitting. The model is trained for 100 epochs with a batch size of 20,480, using the Adam optimizer with a learning rate of 0.001 and weight decay of 0.001 on a single NVIDIA A100 GPU for

Table 1: **Summary of datasets for downstream tasks.**

| Abbreviation | Description | Data Source | Spatial Scale | Sample number | Publicly Available |
|---|---|---|---|---|---|
| Social Tasks | | | | | |
| **POP** | Population density | WorldPop (Bondarenko et al., 2025) | Grid | 194,232 | Y |
| **EDU** | Average education years | Census Year Book (NBS, 2020) | County/District | 2,800 | Y |
| **ELD** | Elderly (65+) population ratio | Census Year Book (NBS, 2020) | County/District | 2,800 | Y |
| **HSR** | Hukou separation ratio | Census Year Book (NBS, 2020) | County/District | 2,800 | Y |
| **CRI** | Crime cases | Zhang et al. (2025) | Grid | 39,932 | Y |
| Economic Tasks | | | | | |
| **NTL** | Nighttime light intensity | MOSAIKS (Rolf et al., 2021) | Point | 23,333 | Y |
| **HOU** | Per capita housing area | Census Year Book (NBS, 2020) | County/District | 2,800 | Y |
| **ENE** | Energy consumption | City statistical yearbooks | City/Prefecture | 283 | N |
| **COS** | Offline consumption amount | Tencent | Grid | 194,052 | N |

one hour. During training, we conduct a grid search over commonly used hyperparameters: learning rate $\in \{1e-4, 2e-4, 5e-4, 1e-3, 2e-3, 5e-3\}$ and weight decay $\in \{1e-3, 1e-2, 1e-1\}$. The LINE embeddings for the nationwide mobility graph were computed on Tencent's Angel distributed graph computing platform (Jiang et al., 2018), utilizing hundreds of computing cores. Graph nodes with more than a billion edges were embedded efficiently within a few hours.

### 4.1.2 DOWNSTREAM EVALUATION

**Downstream tasks and datasets.** To assess our model's generalizbility, we curated a set of 9 downstream tasks as shown in Table 1. These tasks were selected for their diversity and broad representativeness, spanning the two key domains of human-centric scenarios: social and economic. Considering the influence of statistical unit delineation in geospatial analysis, the downstream tasks are designed to span four distinct spatial scales — point, grid, county/district, and city/prefecture — to examine resolution effects in region representation learning. The samples are evenly distributed across the study area, ensuring an evaluation free of geographic bias. Detailed descriptions and spatial distributions of the dataset samples are provided in Appendix A.4. MoRA is trained at the H3 grid level. For point-level tasks, individual point values within each grid are aggregated to align with the grid-level embeddings. For administrative-level tasks, we average embeddings over all grids within each unit and use the resulting mean for regression. This multi-scale evaluation allows us to assess the model's generalizability across varying spatial scales.

**Prediction models.** We implement LightGBM (Ke et al., 2017) as the downstream prediction model. The hyperparameters are lightly tuned, and the average performance on the test set over 10 runs is reported along with the standard deviation. To enhance performance while mitigating the risk of overfitting, different sets of hyperparameters are used for tasks at different levels, considering the variation in sample sizes. For city-level electricity consumption prediction tasks, Ridge regression is adopted due to the limited number of data samples.

### 4.2 PERFORMANCE COMPARISON

**Baseline models.** We benchmark MoRA against the strong existing approaches from two categories:

1. Approaches with publicly available, pre-trained embeddings, which we evaluate directly on our benchmark tasks and datasets. These can be further grouped into: 1) Self-supervised pretraining approaches: AlphaEarth (Brown et al., 2025), SatCLIP (with ViT-16) (Klemmer et al., 2025), GeoCLIP (Vivanco Cepeda et al., 2023), and CSP (Mai et al., 2023a); and 2) analytical position-encoding functions: Sphere2Vec (sphereC+) (Mai et al., 2023b), Siren (Rußwurm et al., 2024), and NeRF (Mildenhall et al., 2021). (The selection is based on methods' superior performance reported in previous work (Wu et al., 2024)). Note that MoRA operates on H3 grids as its spatial unit. To ensure consistency, for the location encoders used for comparison that take 2D coordinates as input, we generate location embeddings for the centroids of the H3 grids for comparison with our grid embeddings. For the pixel-level encoder, i.e., AlphaEarth, the mean of all pixel embeddings within each grid is computed as the baseline. We use grids covering the entirety of China for comparison.

Table 2: **Comparison of model performances on 9 downstream tasks spanning social and economic aspects.** Previous best results are underlined, while best results are highlighted in **bold**. Model performances are measured by $R^2$, and the standard deviations over ten repeated trainings are marked in brackets below. Row $\Delta$ illustrates the percentage difference between Our Method's performance and the best baseline.

| Model | Dim↓ | Social Tasks | | | | | Economic Tasks | | | |
|---|---|---|---|---|---|---|---|---|---|---|
| | | POP | EDU | ELD | HSR | CRI | NTL | HOU | ENE | COS |
| State-of-the-art pretrained Location Encoders | | | | | | | | | | |
| AlphaEarth | 64 | 0.80 | 0.77 | 0.71 | 0.68 | 0.71 | **0.63** | 0.63 | 0.47 | 0.81 |
| | | (0.02) | (0.01) | (0.02) | (0.03) | (0.01) | (0.02) | (0.03) | (0.02) | (0.01) |
| SatCLIP | 256 | 0.52 | 0.63 | 0.68 | 0.74 | 0.39 | 0.33 | 0.66 | -0.07 | 0.44 |
| | | (0.03) | (0.01) | (0.03) | (0.03) | (0.01) | (0.03) | (0.03) | (0.11) | (0.03) |
| GeoCLIP | 512 | 0.41 | 0.66 | 0.66 | 0.69 | 0.32 | 0.24 | 0.65 | 0.11 | 0.32 |
| | | (0.04) | (0.01) | (0.02) | (0.03) | (0.01) | (0.03) | (0.02) | (0.01) | (0.00) |
| CSP | 256 | 0.55 | 0.65 | 0.62 | 0.68 | 0.39 | 0.29 | 0.62 | 0.20 | 0.46 |
| | | (0.03) | (0.02) | (0.03) | (0.03) | (0.01) | (0.03) | (0.02) | (0.03) | (0.03) |
| Sphere2Vec | 96 | 0.40 | 0.56 | 0.62 | 0.61 | 0.34 | 0.26 | 0.57 | -0.01 | 0.35 |
| | | (0.03) | (0.02) | (0.04) | (0.03) | (0.01) | (0.03) | (0.02) | (0.01) | (0.03) |
| NeRF | 96 | 0.40 | 0.63 | 0.61 | 0.61 | 0.33 | 0.24 | 0.56 | 0.03 | 0.33 |
| | | (0.03) | (0.02) | (0.03) | (0.03) | (0.01) | (0.03) | (0.02) | (0.05) | (0.02) |
| Siren | 1,024 | 0.51 | 0.66 | 0.69 | 0.74 | 0.39 | 0.33 | 0.66 | -0.14 | 0.44 |
| | | (0.03) | (0.02) | (0.03) | (0.03) | (0.01) | (0.03) | (0.03) | (0.14) | (0.03) |
| Ours | | | | | | | | | | |
| **MoRA** | 128 | **0.83** | **0.85** | **0.81** | **0.81** | **0.76** | 0.62 | **0.70** | **0.72** | **0.91** |
| | | (0.02) | (0.01) | (0.02) | (0.02) | (0.00) | (0.02) | (0.02) | (0.02) | (0.01) |
| $\Delta$ | | + 4.1% | + 10.5% | + 13.9% | + 9.5% | + 6.9% | -1.7% | +6.1% | +54.5% | +12.1% |

2. Approaches that are not directly applicable, requiring further work to generate embeddings, either by training a published model on our dataset or by sourcing compatible input data to perform inference with their pre-trained weights. Since most existing models from this type were designed and tested at the city scale instead of a national one, we implement all models in Jiangsu Province for a fairer comparison (see Figure 2 for Jiangsu's location). We select HREP (Zhou et al., 2023), ReCP (Li et al., 2024), and ReFound (Xiao et al., 2024) as representative works from each type as baselines. See Appendix A.5 for details of the baselines and our implementation for comparison.

**Comparison results.** In Table 2, we compare MoRA's predictive performance against pre-trained embeddings across a range of tasks and report results by downstream task category. Clearly, MoRA delivers superior performance across all tasks, averaging a 12.9% improvement overall, with 10.8% and 16.0% gains for the social and economic tasks, respectively. Given that human-centric tasks hold great value in practice and typically do not exhibit easily inferable location-dependent properties, an average $R^2$ of 0.78 across nine diverse tasks demonstrates that MoRA effectively captures complex, high-level regional correlations beyond simple geographic proximity, delivering strong practical value. Note that the very slight underperformance on nighttime-light prediction (1.7%) likely stems from implicit correlations and co-variation between common Earth observation features and the target indicator. We further evaluated MoRA's performance in predicting urban carbon emissions, a task spanning natural and economic domains (Appendix A.2). MoRA achieved a superior performance than AlphaEarth, demonstrating that the mobility backbone captures correlations between intensive human activity and environmental externalities that physical-only encoders might overlook.

The superiority of MoRA is consistent across varied spatial scales, achieving the best performance on tasks at city, county, and grid levels. Notably, in the city-level energy consumption task, MoRA demonstrates impressive results with a prediction accuracy of 0.72 in terms of $R^2$. In contrast, the baseline models exhibit $R^2$ values mostly fluctuating around zero, while our approach exceeds the second-best by 54.5%, revealing their limited capacity to capture regional features at coarser scales.

For the second category of approaches, Table 3 shows that MoRA, with a compact 128-dimensional representation, consistently outperforms both baselines, featuring an average gain of 20.9% over the second-best. This underscores MoRA's methodological advantages as a generalizable framework.

Table 3: **Performance comparison between MoRA and methodological baselines in Jiangsu Province ($R^2$).** The $\Delta$ row reports the percentage improvement over the strongest baseline.

| Model | Dim↓ | Social Tasks | | | | | Economic Tasks | | | |
|---|---|---|---|---|---|---|---|---|---|---|
| | | POP | EDU | ELD | HSR | CRI | NTL | HOU | ENE | COS |
| HREP | 128 | 0.57 | 0.55 | 0.24 | 0.49 | 0.68 | 0.34 | 0.42 | 0.38 | 0.73 |
| ReCP | 96 | 0.34 | **0.78** | 0.34 | 0.55 | 0.73 | 0.45 | 0.33 | 0.26 | 0.64 |
| ReFound | 768 | 0.41 | 0.71 | 0.45 | 0.51 | 0.73 | 0.47 | 0.35 | **0.80** | 0.68 |
| MoRA | 128 | **0.68** | 0.75 | **0.55** | **0.81** | **0.74** | **0.55** | **0.53** | 0.73 | **0.85** |
| $\Delta$ | | + 18.0% | - 4.1% | + 22.4% | + 47.2% | + 1.4% | + 16.9% | + 25.7% | - 8.4% | + 15.6% |

**Inference.** For practical deployment, we develop and open-source a distilled version of MoRA that captures core geospatial knowledge and enables direct, coordinate-only inference. Specifically, we train a multi-layer perceptron (MLP) to map continuous geographic coordinates to high-dimensional embeddings, supervised by MoRA's grid-level outputs. This allows practitioners to generate high-quality, privacy-preserving embeddings at arbitrary locations using only geographic coordinates, completely eliminating the need for external datasets and H3-based lookups at inference time. Detailed implementation of the distillation approach is provided in Appendix A.1.

## 4.3 SCALING BEHAVIOR IN GEOSPATIAL REPRESENTATION LEARNING

Echoing scaling behaviors observed in LLMs (Kaplan et al., 2020), we examine how model performance in geospatial representation learning scales with pretraining data size and spatial coverage. Figure 2 shows model performances of three MoRA variants trained on three nested spatial coverages, namely *Jiangsu*, *East China*, and *China*. The spatial extents are shown in Figure 2(r). We conducted three separate experiments with different pretraining data sizes where *Jiangsu* contains 3,904 H3 cells, *East China* contains 28,855 cells and *China* contains 195,574 cells. All three pretrained models are tested in *Jiangsu* only, using evaluation methods described in Section 4.1.2.

We observe that downstream performance generally increases with larger pretraining data for both social and economic tasks (Figure 2, Appendix A.3). The performance gain from *Jiangsu* to *East China* is especially large, while the increase from *East China* to *China* is relatively marginal. External training information can be useful for internal predictions. The marginal benefits of increasing training data size gradually decline. The scaling behavior in geospatial representation learning is consistent with findings in language and vision domains, where model performance improves with scale (Henighan et al., 2020). Results emphasize the value and necessity of geospatial representation learning at scale. Even for small-scale predictions, an increase in training data size and spatial coverage will introduce relevant information and improve downstream predictive performances.

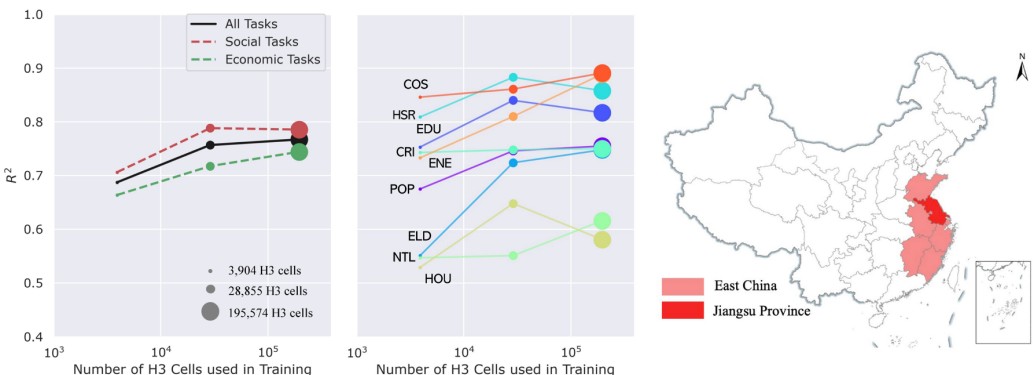

Figure 2: **Scaling behavior of downstream task performance with pretraining data size and spatial coverage.** Map on the right illustrates the spatial extent of Jiangsu Province, East China, and the entire China, respectively covering 3,904, 28,855, and 195,574 H3 cells used for the pretraining. Left figure illustrates average $R^2$ values for all tasks, social tasks, and economic tasks respectively, while the middle figure illustrates task-specific scaling behaviors. Detailed numbers in Appendix A.3.

## 4.4 ABLATION AND SENSITIVITY STUDY

Figure 3: **Model comparison results for ablation studies and sensitivity analysis.**

**Ablation study on anchoring modality.** We frame mobility as the backbone for aligning other modalities and justify this design theoretically. Here, we empirically evaluate its unique advantages over other anchoring choices. As shown in Figure 3(a), mobility-as-backbone significantly outperforms alternative anchor modalities across both task types, achieving an average of 16.8% gain over the second-best (the demo-anchored structure), validating its superiority.

**Module-wise ablation study.** Figure 3(b) shows the performance of multiple model variants on downstream tasks. The columns labeled **w/o CLIP** indicate that removing the CLIP-based alignment module and leaving only the mobility modality leads to an average performance drop of 6.5%. Second, mobility forms the backbone of **MoRA**, with graph information injected at two stages. We assess each by ablating them separately. Removing the graph encoder and substituting a simple MLP (w/o Mob Graph Encoder) causes the largest drop, indicating the GCN captures non-local patterns via relational structure. Replacing LINE initialization with random values (**w/o LINE Initialization**) has a smaller impact, suggesting that the graph structure is more informative than pre-encoded embeddings. However, a consistent 1–2% decline shows that LINE still provides complementary value. We further replace the mobility graph with a simple neighborhood graph in which edges connect only directly adjacent cells **(MoRA(neighborhood graph))**. This leads to a 12.2% drop in average performance, highlighting the mobility graph's ability to capture long-range spatial correlations, i.e., "functional adjacency", that are absent in graphs based solely on direct neighborhood adjacency.

**Ablation on auxiliary modalities.** We ablate each of the three auxiliary modalities, i.e., POIs, satellite imagery, and demographics, to assess their contributions to the model performance on human-centric inference. As shown in Table 4, the removal of deomographics information consistenly results in the largest performance decrease across nearly all tasks, demonstrating its unique advantages in capturinng human-related features, in line with the findings of Wen & Zhou (2024). Images and POIs offer essential additional information, boosting performance across both types of tasks. Overall, MoRA achieves the best performance in every domain and downstream task, validating the effectiveness of its multimodal fusion approach.

Table 4: Ablation on auxiliary modalities in mobility-as-backbone fusion ($R^2$).

| | Social Tasks | | | | | Economic Tasks | | | | AVG $R^2$ ↑ |
|---|---|---|---|---|---|---|---|---|---|---|
| | **POP** | **EDU** | **ELD** | **HSR** | **CRI** | **NTL** | **HOU** | **ENE** | **COS** | |
| w/o POI | 0.835 | 0.835 | 0.796 | 0.800 | 0.755 | 0.590 | 0.702 | 0.680 | 0.895 | 0.765 |
| w/o Image | 0.832 | 0.841 | 0.791 | 0.803 | 0.755 | 0.599 | 0.673 | 0.665 | 0.900 | 0.762 |
| w/o Demo | 0.824 | 0.827 | 0.765 | 0.793 | 0.753 | 0.605 | 0.685 | 0.590 | 0.893 | 0.748 |
| **MoRA** | 0.829 | 0.853 | 0.812 | 0.810 | 0.759 | 0.615 | 0.703 | 0.719 | 0.909 | 0.779 |

**Sensitivity to graph sampling.** Given the mobility graph encoder's importance, we analyze sensitivity to neighbor sampling strategy and ratio (Figure 3(c)). We compare our *top-k* sampling approach with random sampling, which retains links randomly based on a percentage threshold instead of depending on link weights. The results suggest that *top-k* sampling shows minimal performance variation across ratios (top 10% vs. top 30%). Random sampling, by tending to include nodes from the "long-tail", introduces noise and generally underperforms top-k at the same ratios. As a result, random sampling generally underperforms *top-k* sampling at equivalent ratios. Yet as the random ratio increases, more generalized patterns offset the noise, leading to improved performance. Overall, MoRA balances accuracy and efficiency across graph configurations.

**Sensitivity to H3 grid resolution.** We extend our study at H3 resolution level 6 to its adjacent levels, 5 and 7, to evaluate the model's performance across different spatial scales. Compared to level 6 H3 cells, which cover an average area of 36.13 km², cells at levels 5 and 7 cover approximately 253.61 km² and 5.18 km², respectively. The East China region is selected for testing to facilitate ease of operation without losing generalizability. We reconstruct the training dataset for levels 5 and 7 from scratch, including forming trajectory-based mobility graphs, gathering satellite imagery and POIs for encodings, and processing demographic distributions data. Three separate experiments are conducted, with corresponding sets of downstream data processed for evaluation. Since cities in East China are limited in quantity, we choose Ridge Regression for energy consumption prediction across all levels to ensure consistency. For the remaining tasks, LightGBM is adopted. Table 5 reports model performance with varying spatial resolutions. Overall, performance generally improves as spatial granularity increases, though the rationale varies by the level of the downstream tasks. For grid-level predictions, finer resolution can enhance localized prediction. However, an exception is observed with crime prediction, which follows a reversed pattern. This might be explained by the fact that crime becomes more heterogeneous and exhibits greater variations at finer scales. At the county and city levels, where aggregation smooths out local noise and highlights more significant features, performance improves with grid granularity.

Table 5: Sensitivity of model performance using different grid resolutions ($R^2$).

| | Grid-level | | | | County-level | | | | City-level |
|---|---|---|---|---|---|---|---|---|---|
| | **NTL** | **COS** | **CRI** | **POP** | **EDU** | **ELD** | **HOU** | **HSR** | **ELE** |
| H3 lv5 | 0.462 | 0.881 | 0.838 | 0.679 | 0.645 | 0.668 | 0.611 | 0.830 | 0.659 |
| H3 lv6 | 0.632 | 0.901 | 0.764 | 0.817 | 0.790 | 0.698 | 0.665 | 0.835 | 0.668 |
| H3 lv7 | 0.741 | 0.934 | 0.634 | 0.836 | 0.821 | 0.754 | 0.700 | 0.872 | 0.711 |

## 5 CONCLUSION AND DISCUSSION

This work introduces MoRA, a mobility-as-the-backbone framework for geospatial representation learning. Centered around a billion-scale mobility graph and enriched with multimodal alignment (including POIs, remote sensing imagery, and tabular data), MoRA learns unified region embeddings that generalize across diverse downstream tasks. Extensive experiments on 9 socio-economic tasks demonstrate MoRA's consistent outperformance of state-of-the-art methods. Moreover, our findings echo the scaling laws observed in other domains: expanding training data's spatial coverage from local to national scales significantly enhances both representation quality and task performance.

Despite strong empirical results, limitations remain. The scale issue in spatial representation learning is not fully resolved, despite our sensitivity analysis indicating the robustness of MoRA across spatial scales. Second, although MoRA itself as a framework is dataset-agnostic and can be adapted to alternative, publicly available movement data sources (e.g., Internet data or mobile phone data), high-quality human mobility data may be difficult to obtain in some regions. In such cases, we suggest a neighborhood-based variant of our model as a viable and effective alternative that produces high-quality geospatial representation.

## ACKNOWLEDGEMENTS

This work was supported by National Key R&D Program of China (2022YFB3903704). The authors would like to thank Wei Qi for providing the China Statistical Yearbook data and administrative boundary data.

## ETHICS STATEMENT

This work adheres to the ICLR Code of Ethics. Recognizing the privacy considerations associated with human mobility data, we rigorously minimize associated risks through a four-stage **anonymization-aggregation-learning-distillation** pipeline:

1. Anonymization: The mobility data is derived from the locations of offline stores. Grids are linked based on observed mobility flows between stores in different regions within a week, and these links are aggregated into an inter-grid network. No individual-level information is retained.

2. Aggregation: Raw mobility data is aggregated at a much coarser grid level, ensuring that no precise geographic coordinates are recorded.

3. Representation Learning: Embeddings inherently generalize patterns.

4. Model Distillation: We share a distilled model, ensuring that reconstructing the original data or tracing back to specific individuals is practically impossible.

This aligns with established privacy-preserving practices in mobility research. The disclosed embeddings contain no personally identifiable information, and our distillation adds an additional layer of critical protection.

## REPRODUCIBILITY STATEMENT

We have made every effort to ensure a high level of reproducibility of our work. Specifically, we:

- Release the complete codebase at https://github.com/ylzhouchris/MoRA, including data preprocessing pipelines, model architecture, and code for downstream task evaluation.

- Describe the training implementation in detail in Section 4.1, including training steps, hardware details, model hyperparameters and so on.

- Make the pretrained model weights available by open-souring a high-performance distilled version of our model, enabling researchers to immediately benefit from MoRA's capabilities via simple, direct queries.

- Establish and release benchmark downstream dataset, accompanied by detailed descriptions of task properties and visualizations of their geographical distributions in Appendix A.4.

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

# A  APPENDIX

## A.1  DETAILED IMPLEMENTATION OF THE DISTILLATION

For model distillation, we fit a surrogate Multi-Layer Perceptron (MLP) network to estimate pretrained region embeddings from MoRA. For each H3 cell, the input for the MLP is a 1024-d vector, which is a SIREN encoding (Rußwurm et al., 2024) of the centroid coordinate of the cell. The output for the MLP is a 128-d vector from MoRA. The MLP is configured with 8 hidden layers, with a dimension of 512 each. The model is trained with an MSE loss function to minimize the deviation of estimated embeddings from our true embeddings. The learning rate is set to 0.005 for 5000 epochs. Final training loss is 0.0857. The downstream performance of the distilled model is shown in Table 6. Note that the performance of the distilled model (MoRA$_d$) tends to improve as the number of MLP layers and the size of the hidden dimensions increase. We here demonstrate the distillation effectiveness preliminarily using the model size mentioned above.

Table 6: Performance comparison between MoRA$_d$ and baselines.

|  | **Social Tasks** | | | | | **Economic Tasks** | | | |
|  | **POP** | **EDU** | **ELD** | **HSR** | **CRI** | **NTL** | **HOU** | **ENE** | **COS** |
|---|---|---|---|---|---|---|---|---|---|
| SatCLIP | 0.523 | 0.633 | 0.676 | 0.735 | 0.391 | 0.334 | 0.655 | -0.069 | 0.443 |
| MoRA | 0.829 | 0.853 | 0.812 | 0.810 | 0.759 | 0.615 | 0.703 | 0.719 | 0.909 |
| MoRA$_d$ | 0.730 | 0.835 | 0.804 | 0.816 | 0.580 | 0.517 | 0.693 | 0.742 | 0.729 |

## A.2  GENERALIZATION TO URBAN CARBON EMISSION PREDICTION

While MoRA is primarily optimized for socio-economic contexts, its utility extends to environmental domains where human activity serves as a primary driver. We evaluated MoRA against AlphaEarth in predicting nationwide spatio-temporal urban carbon emissions (2020–2024) using a high-resolution satellite-derived dataset (Fan et al., 2025). To account for the temporal dynamics of emissions, we augmented the static 128-dimensional representations with two cyclical features: time of year and time of day. In urban areas, MoRA ($R^2 = 0.612$) outperforms AlphaEarth ($R^2 = 0.605$). This result suggests that a mobility-anchored backbone effectively captures the intensive human activity patterns, such as road traffic, that dictate urban emission externalities, which surface-level physical encoders may partially overlook.

## A.3  RESULTS TABLE FOR THE SCALING BEHAVIOR

Table 7 reports task-wise performance for the scaling experiments described in Section 4.3 , which also serves as the source data for Figure 2.

Table 7: Downstream task performance (R²) under model variants in scaling behavior.

|  | **Jiangsu** | **East China** | **China** |
|---|---|---|---|
| **Social** | | | |
| POP | 0.675 | 0.746 | 0.755 |
| EDU | 0.753 | 0.840 | 0.817 |
| ELD | 0.551 | 0.724 | 0.748 |
| HSR | 0.809 | 0.883 | 0.858 |
| CRI | 0.743 | 0.748 | 0.750 |
| **Economic** | | | |
| NTL | 0.547 | 0.551 | 0.616 |
| HOU | 0.529 | 0.648 | 0.581 |
| ENE | 0.733 | 0.810 | 0.889 |
| COS | 0.846 | 0.861 | 0.891 |

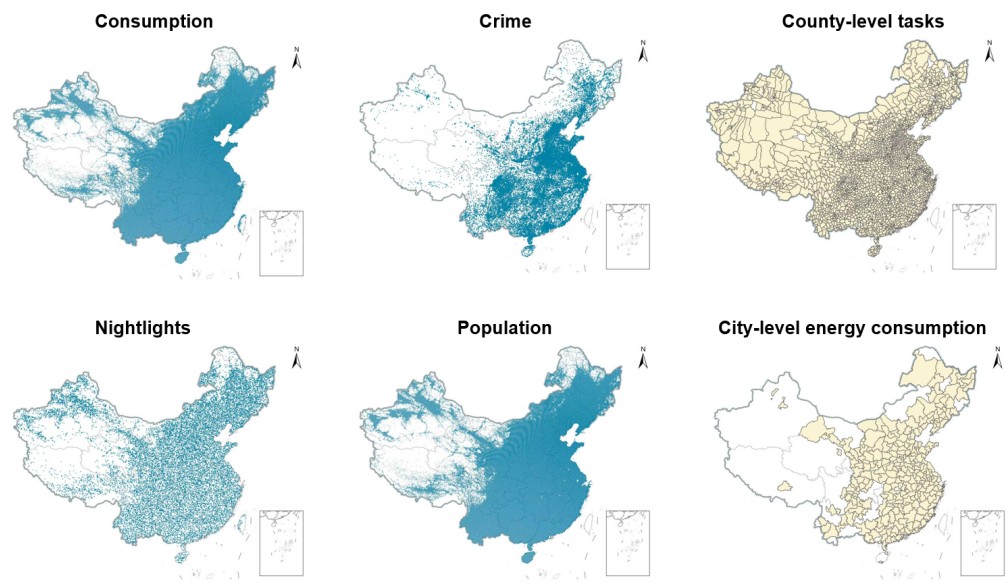

Figure 4: The geographical distribution of various downstream dataset samples.

## A.4    DETAILS ON DOWNSTREAM DATASETS AND SPATIAL DISTRIBUTION

The data sources summarized in Table 1 are primarily drawn from WorldPop (Bondarenko et al., 2020), MOSAIKS (Rolf et al., 2021), and the 2020 China Census Year Book from the National Bureau of Statistics of China (NBS, 2020). Additionally, ENE (energy consumption) is compiled from various local statistical yearbooks. The total energy consumption of one city/prefecture is computed by converting different forms of energy usage into tonnes of standard coal equivalent. We also include an unavailable-to-the-public dataset, COS (offline consumption amount) from Tencent, to evaluate whether the embeddings can predict regional consumption indices. For the crime dataset, using the original data from (Zhang et al., 2025), we aggregate cases over a five-year interval, add 1 and apply a log transform to the labels.

Figure 4 depicts the geographical distribution of the downstream task datasets. Overall, samples are widely spread across China. Note, however, that the west contains vast plateaus and deserts (e.g., the Tibetan Plateau) with minimal human activity, leading to sparse coverage for several tasks. In regions with active human presence, samples are distributed fairly evenly. Importantly, these maps reflect only the task-specific data availability—some sparsity stems from the nature of the tasks or their collection processes—and is independent of our region embeddings, which are uniformly distributed across populated areas of China.

## A.5    BASELINES

We benchmark our framework against two types of baselines: (i) feature-based extractors that enable direct, coordinate-based, training-free embedding queries through pretraining or analytical transformation functions, and (ii) methods that require either retraining or the preparation of a strictly matching input dataset. Since the first category aligns closely with our approach and is covered in details in *Related Works*, the discussion here focuses on the second category. A notable line of work in this category adopts graph-based multi-view representation learning: for each view (e.g., mobility, POI), a graph is constructed based on attribute similarity, and the resulting node embeddings are fused (e.g., via attention mechanisms) to produce unified region representations. Contrastive-based methods—mostly formulated between two views or modalities of the data—are also widely used for city-scale region representation learning. Besides, there are early attempts for building region representation fundation model through multi-city pretraining or distillation from LLMs and VLMs. Accordingly, we select three representative state-of-the-art works from each category as baselines:

- **HREP** constructs a heterogeneous graph from three types of regional data, namely, human mobility, POIs and geographic neighboring relationship, and learns region embeddings using a relation-aware GCN and attention-based cross-view fusion.

- **ReCP** leverages views from POI and human mobility data, applies contrastive learning at both intra-view and inter-view levels to enhance the cross-view consistency, and finally fuse the two views via concatenation.

- **ReFound** integrates multi-source grospatial data through a transformer-based structure and distills knowledge from language, visual foundation models joinlt to enhance its generalization capabilities. The model is pretrained in five major cities in China to acquire general geospsatial knowledge.

The baselines for HREP and ReCP require re-training, whereas ReFound provides pre-trained weights for inference. However, because the datasets for both input features and downstream evaluation are unavailable for all three models, we adapt their codebases and released weights to our datasets in the same study area. Specifically, we find and align required data with each model's input format, re-train HREP and ReCP, and run inference with ReFound.

## A.6 The Use of Large Language Models (LLMs)

We employed a large language model (LLM) exclusively for language polishing, including grammar correction, stylistic refinement, and improvements to clarity and readability. The LLM did not contribute to the conception of the study, data collection, model design, statistical analysis, or interpretation of results, nor did it generate figures, tables, or substantive content. All methodological choices, analyses, and conclusions were conceived, executed, and validated by the authors. Drafts edited with the LLM were subsequently reviewed line by line by the authors to ensure technical accuracy, fidelity to the original meaning, and alignment with ethical and scholarly standards. The authors bear full responsibility for all scientific content presented in this work.

