# OpenReview forum: "MoRA: Mobility as the Backbone for Geospatial Representation Learning at Scale"
_ICLR.cc/2026/Conference — ICLR 2026 Poster_

### Official Review · Reviewer_L9Si · 2025-10-24

**Soundness:** 3
**Presentation:** 3
**Contribution:** 2
**Rating:** 4
**Confidence:** 5

**Summary:**

MoRA presents a human-centric geospatial representation paradigm that uses billion-scale mobility graphs as the backbone: it encodes H3 grid cells into 128-D vectors with GNNs, aligns satellite imagery, POIs and demographics via contrastive learning, boosts downstream socio-economic tasks by 12.9% on average, demonstrates the first scaling laws in GeoAI, and releases a privacy-preserving distilled model that maps coordinates directly to high-quality embeddings.

**Strengths:**

MoRA leverages a billion-edge mobility graph as its backbone, aligns imagery, POIs and demographics into a unified 128-D space via contrastive learning, delivers a 12.9% average gain across nine nationwide socio-economic tasks, demonstrates the first scaling laws in GeoAI, and releases a privacy-preserving distilled model that infers embeddings from coordinates alone.

**Weaknesses:**

1. The design of downstream evaluation does not align with main stream methodology.
Why utilize LightGBM as the downstream prediction model instead of a linear probing method that is employed by many existing works [1,2,3,4]

2. The coverage of dataset is limited. MoRA only utilized dataset from China. How to ensure the generalization of the induced conclusion?
It is suggested to compare more cities in othe counties.


3. The technique contribution is limited. Actually, combining mobility with POI and Satellite images is a common practice in region representation learning.
What MoRA has done is move this paradigm into location embedding, while the requirement of multiple different kinds of data modality also diminish the generalization and practicability. That is why previous location embedding work [1,2,3] applied global application while this work only limited on China data.

4. I am confused of the inference practice. The paper states that during inference, it still needs 3 kinds of data (POI, demographic, satellite images) for prediction. If so, what is the meaning of contrastive learning in pretraining stage? The CLIP-based contrastive learning should inject the information of POI, image, demographics data modality into mobility backbone through contrastive alignment in pretraining. In inference, the common practice is to only utilize the pretrained mobility backbone.

---
[1] Klemmer, Konstantin, et al. "Satclip: Global, general-purpose location embeddings with satellite imagery." Proceedings of the AAAI Conference on Artificial Intelligence. Vol. 39. No. 4. 2025.

[2] Hao, Xixuan, et al. "Nature makes no leaps: Building continuous location embeddings with satellite imagery from the web." Proceedings of the ACM on Web Conference 2025. 2025.

[3] Vivanco Cepeda, Vicente, Gaurav Kumar Nayak, and Mubarak Shah. "Geoclip: Clip-inspired alignment between locations and images for effective worldwide geo-localization." Advances in Neural Information Processing Systems 36 (2023): 8690-8701.

[4] Hao, Xixuan, et al. "Urbanvlp: Multi-granularity vision-language pretraining for urban socioeconomic indicator prediction." Proceedings of the AAAI Conference on Artificial Intelligence. Vol. 39. No. 27. 2025.

**Questions:**

See weakness.

---

> ### Author Response · Authors · 2025-11-21
> **Response to Reviewer L9Si - Part 1**
>
> We thank the reviewer for the comments, and we address your concerns as follows:
>
> >Q1. The design of downstream evaluation does not align with main stream methodology. Why utilize LightGBM as the downstream prediction model instead of a linear probing method that is employed by many existing works [1,2,3,4]
>
> **A1:** We thank the reviewer for this comment regarding the evaluation protocol. While **linear probing** (simple linear regression) is indeed a frequently used model for downstream evaluation, a variety of methods have also been adopted in the existing literature. In fact, among the works cited by the reviewer, [1,2,4] utilize MLPs and [3] utilize similarity-based retrieval, all being non-linear methods. The criteria for choosing the appropriate downstream evaluation model, as the authors believe, should be 1) the downstream evaluation model can reveal the optimal prediction potential of the learned representations, and 2) the evaluation model is applied consistently across all baselines to ensure a fair comparison.
>
> We specifically selected LightGBM for two key reasons:
> 1. **Superior Prediction Performance:** Our preliminary tests demonstrated that LightGBM yielded superior prediction performances compared to linear models.
> 2. **Robustness and Efficiency:** Tree-based models are widely recognized for their robustness, strong performance, and relative insensitivity to hyperparameters [5,6]. LightGBM's high robustness and computational efficiency are particularly valuable for handling our multi-task, nation-scale datasets.
>
> Crucially, this performance uplift was observed across **all** baselines, not just our own. Therefore, using LightGBM ensures a fair comparison while allowing all methods to demonstrate their optimal potential.
>
>
> >Q2. The coverage of dataset is limited. MoRA only utilized dataset from China. How to ensure the generalization of the induced conclusion? It is suggested to compare more cities in other counties.
>
> **A2:** We appreciate the reviewer raising the point about generalizability. While our study utilizes data from China, we argue that our methodology and findings have a high level of generalizability because the vast internal diversity of our study area acts as a **strong proxy for global variation**. This region encompasses immense differences in economic development, population density, and topography. To empirically validate this, we conducted a rigorous cross-region evaluation across three distinct provinces—representing highly urbanized, dense (Jiangsu and Guangdong), and sparsely populated, mixed-topography regions (Sichuan). As quantified by the metrics in our tables, our method maintains **consistent performance** across these fundamentally different geographic contexts ($\Delta$< 5 \% relative to national benchmarks), demonstrating robustness to the kinds of heterogeneity expected between different countries. Furthermore, our model is based on universal principles about human mobility and location relations, not country-specific priors.
>
> Table Ⅰ. Descriptive statistics across regions.
> ||**Population density (people/km²)**|**Checkin counts Mean**|**POI counts Mean**|**Population density Moran's I**|**Checkin counts Moran's I**|**POI counts Moran's I**|
> |-|-|-|-|-|-|-|
> |Sichuan|163|521,749|712|0.395|0.410|0.346|
> |Guangdong|493|2,202,197|3,076|0.458|0.512|0.448|
> |Jiangsu|616|1,761,805|2,693|0.231|0.332|0.239|
>
>
> Table Ⅱ. Regional validation results (R²).
> ||**POP**|**EDU**|**ELD**|**HSR**|**CRI**|**NTL**|**HOU**|**COS**|**Avg.**|
> |-|-|-|-|-|-|-|-|-|-|
> |Sichuan|0.716|0.751|0.881|0.755|0.765|0.465|0.467|0.865|0.708
> |Guangdong|0.847|0.772|0.740|0.800|0.824|0.707|0.599|0.888|0.772
> |Jiangsu|0.758|0.818|0.748|0.860|0.749|0.617|0.581|0.891|0.753

---

> ### Author Response · Authors · 2025-11-21
> **Response to Reviewer L9Si - Part 2**
>
> >Q3:  The technique contribution is limited. Actually, combining mobility with POI and Satellite images is a common practice in region representation learning. What MoRA has done is move this paradigm into location embedding, while the requirement of multiple different kinds of data modality also diminish the generalization and practicability. That is why previous location embedding work [1,2,3] applied global application while this work only limited on China data.
>
> **A3:** We thank the reviewer for this comment. We respectfully clarify that our primary contribution lies not merely in the use of multimodal data, but also in the novel way we conceptualize and integrate these modalities.
>
> We treat grid cells as **"spatial tokens"** and mobility patterns as **"sentences"** that provide semantic context. We contend that such **structural innovation** is often more pivotal than data selection. Mirroring the success of **ResNet** and **Transformers**, which revolutionized deep learning by capturing **long-range dependencies**, our framework breaks the constraints of local neighborhoods to enable information propagation across vast geographic distances.This focus on **inter-cell functional relationships** fundamentally distinguishes our methodology from prior works [1,2,3], which are typically limited to local aggregations.
>
> Regarding generalization, we have successfully deployed this framework at a massive scale, open-sourcing embeddings for the entire territory of China (**~200,000 grid cells and 1.2 billion mobility links**). While the current implementation utilizes comprehensive data available in China, the underlying methodology is universal. Extending this framework to other countries is a prioritized next step in our roadmap.
>
>
> >Q4:  I am confused of the inference practice. The paper states that during inference, it still needs 3 kinds of data (POI, demographic, satellite images) for prediction. If so, what is the meaning of contrastive learning in pretraining stage? The CLIP-based contrastive learning should inject the information of POI, image, demographics data modality into mobility backbone through contrastive alignment in pretraining. In inference, the common practice is to only utilize the pretrained mobility backbone.
>
> **A4:** Thank you for raising this point. We confirm that **our framework does not require POI, demographic, or satellite image data for final inference.** The purpose of the CLIP-based contrastive pretraining is exactly to **inject** the rich, multi-modal information into the mobility-based representation space, allowing the mobility backbone to be used alone during inference. We follow the common practice of only utilizing the pretrained mobility backbone for direct queries. In fact, we developed and open-sourced a highly practical, distilled version of the model that enables direct embedding queries using only geographic coordinates. We acknowledge that the original manuscript may have caused confusion and have added a dedicated subsection (Lines 388-394) to the revised manuscript to clearly elaborate on this deployment and inference strategy.
>
> ---
> [1] Klemmer, Konstantin, et al. "Satclip: Global, general-purpose location embeddings with satellite imagery." AAAI. 2025. \
> [2] Hao, Xixuan, et al. "Nature makes no leaps: Building continuous location embeddings with satellite imagery from the web." Proceedings of the ACM on Web Conference. 2025. \
> [3] Vivanco Cepeda, Vicente, Gaurav Kumar Nayak, and Mubarak Shah. "Geoclip: Clip-inspired alignment between locations and images for effective worldwide geo-localization." NeurIPS. 2023. \
> [4] Hao, Xixuan, et al. "Urbanvlp: Multi-granularity vision-language pretraining for urban socioeconomic indicator prediction." AAAI. 2025. \
> [5] Agarwal, Mohit, et al. "General geospatial inference with a population dynamics foundation model." arXiv preprint. 2024. \
> [6] Grinsztajn, Léo, Edouard Oyallon, and Gaël Varoquaux. "Why do tree-based models still outperform deep learning on typical tabular data?" NeurIPS. 2022.

---

### Official Review · Reviewer_adgb · 2025-10-26

**Soundness:** 3
**Presentation:** 3
**Contribution:** 3
**Rating:** 6
**Confidence:** 4

**Summary:**

SUMMARY: The authors propose a new geospatial location representation learning framework, leveraging multimodal input data, namely imagery, POI data and demographic information (the latter two available as text). The most unique aspect about this work is that instead of leveraging raw geographic context (e.g. pure coordinates, as used in SatCLIP or GeoCLIP), the authors build a spatial graph between locations based on whether locations are associated with each other in the WeChat Payment system. While this obviously assumes a lot of prior knowledge, it is an interesting approach to explore specifically for geo representation learning in urban areas, where we can assume such data to be available. This gives their whole approach a strongly informed "geographic prior". The authors test their geo embeddings on a set of different socio-economic tasks, showing impressive performance and outperforming recent embeddings such as GeoCLIP or AlphaEarth.

**Strengths:**

STRENGTHS:

- I like the framing of two paradigms for geo representation learning, the human and EO centric ones. Not sure I necessarily agree but this is an interesting framing to motivate the paper! The argument about the relative nature of a "place" (i.e. a location being majorly defined by its relationship with other locations) is thought provoking and interesting. Overall, really great motivation section!

- Exploring scaling laws for geospatial representation learning is a critical research challenge, e.g. also mentioned here [1]. The authors do that a little bit (though I would have liked to see more of that).

- I like that the authors clearly define distinctions from PDFM, which is conceptually closest.

**Weaknesses:**

- I would love to have seen a comparison to a simple neighborhood graph; the way understand it, edges between cells/nodes are based on a-priori known WeChat Pay interactions. This is great if you have access, but what if you didn't have that data? How would this method perform if you simply built your graph based on direct neighborhood/adjacency of cells? How would the model perform then? This seems like a crucial ablation missing, unless I am missing it. This also would allow the authors to compare their approach to PDFM, which is only available for the US - and which would be an important comparison.

- In the abstract, the authors say that "While Earth observation paradigms excel at depicting locations in their physical states, we claim that a location’s comprehensive “meaning” is better grounded in its internal human activity pattern". I don't want to start a philosophical discussion about "place", but this sentence feels a bit strong / definitive. Human activity might give a location "meaning" relevant to some applications - especially human centric ones, but might be irrelevant for other, e.g. natural processes.

- Given that your paper basically argues for a geographic prior for SSL geo-representations (in your case based on payment data), I think it would be important to link your work back to the origins of work on "geographic priors", especially [2].

- I don't like averaging over R2 values of different tasks in Tab 2; this does not seem very rigorous and I'd recommend to remove that column.

- Fig 2: sizes of legend circles don't match the circle sizes in the plot

**Questions:**

Overall this is an interesting, well motivated paper. I like both the methodological innovations and the experiments, though there could be more details and an important ablation is missing. To help with my understanding and the final assessment, I would ask the authors to address my concerns and questions outlined in the "Weaknesses" sections.

References:

[1] Rolf, Esther, et al. "Mission Critical--Satellite Data is a Distinct Modality in Machine Learning." arXiv preprint arXiv:2402.01444 (2024).
[2] Mac Aodha, Oisin, Elijah Cole, and Pietro Perona. "Presence-only geographical priors for fine-grained image classification." Proceedings of the IEEE/CVF International Conference on Computer Vision. 2019.

---

> ### Author Response · Authors · 2025-11-20
> **Response to Reviewer adgb**
>
> We are highly appreciative of the reviewer for recognizing the innovation of our method, the model's impressive performance, and the value of our experiments on scaling laws. We are particularly encouraged that the reviewer appreciates the paper's overall framing and motivation. Below we address the constructive comments and suggestions one by one.
>
> >Q1. I would love to have seen a comparison to a simple neighborhood graph; the way understand it, edges between cells/nodes are based on a-priori known WeChat Pay interactions. This is great if you have access, but what if you didn't have that data? How would this method perform if you simply built your graph based on direct neighborhood/adjacency of cells? How would the model perform then? This seems like a crucial ablation missing, unless I am missing it. This also would allow the authors to compare their approach to PDFM, which is only available for the US - and which would be an important comparison.
>
> **A1:** We thank the reviewer for raising this important point.
>
> We agree that a comparison against a simple neighborhood graph would be informative and valuable. We have conducted an additional experiment where we replaced the mobility graph with a simple neighborhood graph (based on spatial adjacency), keeping the three auxiliary modalities unchanged.
>
> Table I shows that MoRA outperforms its neighborhood-graph-based variant across all tasks. This superior performance is directly attributed to the mobility graph's ability to capture **long-range spatial correlations** (often referred to as "functional adjacency"), which are inherently missed by graphs based solely on direct physical adjacency.
>
> Nevertheless, the variant using the neighborhood graph still achieves strong performance, highlighting both the importance of local spatial context and the effectiveness of our core multimodal alignment strategy. This indicates that the neighborhood-based variant may serve as **a robust and effective alternative** in deployment scenarios where proprietary mobility data is unavailable, strengthening the generalizability of the MoRA framework. We could not compare with PDFM, as the official code and some core datasets (e.g., search trend data) are not publicly available. We will incorporate this comparison once the code is released.
>
> We thank the reviewer again for bringing this important ablation to our attention, as it significantly enhances the completeness of our work. We have updated the results in the ablation section to address this point (Lines 464-466 and Figure 3).
>
>
> Table Ⅰ. Ablation on relational graph.  (R²).
> ||**POP**|**EDU**|**ELD**|**HSR**|**CRI**|**NTL**|**HOU**|**ENE**|**COS**|
> |-|-|-|-|-|-|-|-|-|-|
> |MoRA (neighborhood graph)|0.81|0.82|0.77|0.72|0.61|0.57|0.66|0.51|0.73
> |**MoRA**|**0.83**|**0.85**|**0.81**|**0.81**|**0.76**|**0.62**|**0.70**|**0.72**|**0.91**|
>
>
> >Q2. In the abstract, the authors say that "While Earth observation paradigms excel at depicting locations in their physical states, we claim that a location’s comprehensive “meaning” is better grounded in its internal human activity pattern". I don't want to start a philosophical discussion about "place", but this sentence feels a bit strong / definitive. Human activity might give a location "meaning" relevant to some applications - especially human centric ones, but might be irrelevant for other, e.g. natural processes.
>
> **A2:** We thank the reviewer for pointing this out sharply. We fully understand the difference between "location" and "place", and indeed, the original sentence is a bit definitive. We have revised the sentence as follows,
>
> *"While Earth observation paradigms excel at depicting locations in their physical states, we propose that a location’s full characterization requires grounding in both its physical attributes and its internal human activity pattern, the latter being particularly crucial for understanding its **human-centric functions**."*
>
> We hope this sentence provides a more precise argument.
>
> >Q3:  Given that your paper basically argues for a geographic prior for SSL geo-representations (in your case based on payment data), I think it would be important to link your work back to the origins of work on "geographic priors", especially [2].
>
> **A3:** We thank the reviewer for the insightful reminder and have revised the manuscript to explicitly discuss this connection with "geographic priors" (Lines 44-46).
>
> >Q4: I don't like averaging over R2 values of different tasks in Tab 2; this does not seem very rigorous and I'd recommend to remove that column.
>
> **A4:**  We agree with the reviewer's suggestion and have removed the average column from Table II in the revised manuscript.
>
> >Q5: Fig 2: sizes of legend circles don't match the circle sizes in the plot
>
> **A5:**  Thank you for pointing this out. We have corrected it in Figure 2 of the revised manuscript.

---

### Official Review · Reviewer_Ay9J · 2025-10-30

**Soundness:** 3
**Presentation:** 3
**Contribution:** 3
**Rating:** 6
**Confidence:** 3

**Summary:**

### Problem

The authors tackle the problem geospatial intelligence. Specifically, the seek to convert locations into useful vector embeddings to support tasks such as predicting socio-economic indicators or urban dynamics. They argue existing approaches fall into two silos:
	1.	Earth-observation-centric: learning from satellite imagery and physical features.
	2.	Human-activity-centric: learning from mobility and demographic patterns.

These paradigms rarely integrate well, and current models miss long-range functional relationships between places (e.g., commuter flows) and lack scalable benchmarks. The field also lacks foundation-style models trained at national or global scale. The authors argue that (2) Human-centric representations are fundamental the semantics of place and are a first class citizen in their solution


### Solution

The authors propose MoRA, a framework that leverages mobility as the backbone for geospatial representation learning at scale. Their method has the following properties:
	•	Aggregates real mobility flows across millions of locations and billions of edges.
	•	Builds a nationwide mobility graph using H3 spatial grids.
	•	Learns human-centric location embeddings with a graph neural network
	•	Aligns mobility with three auxiliary modalities using asymmetric CLIP-style contrastive learning.
	•	Demonstrates scaling laws in geospatial representation learning.
	•	Releases a benchmark of 9 socio-economic tasks and a distilled privacy-preserving model.

MoRA consistently outperforms state-of-the-art representations across prediction tasks.

**Strengths:**

- Strong empirical results. Mora outperforms all baselines across several experiments
- Ablation studies validates the additional complexity of their method
- Experiments are done across several training runs, informing reproducibility. Means and standard deviations are provided.
- The introduction of the new geospatial benchmark is noteworthy
- The experiments around scaling laws are useful for the GeoAI community

**Weaknesses:**

- Heavy reliance on proprietary mobility data:
    - This causes concerns related to reproducibility. I don't believe this resource is publicly available
    - What about rural areas where human activity is sparse?
- Single country mobility data: The mobility data is unique to china, casting fairly large doubts on the generalizability of the method. For example, does this method generalize to Europe or the US?
- The new benchmark primarily describes human-centric socio-economic tasks. It would make sense that mobility data would be more useful here (as demonstrated in their experiments). Other tasks such as land cover classification would likely see better results under the more Earth-observation-centric representation learning strategy. Unfortunately, this trade-off is left unexplored.

**Questions:**

- My main question is about generalization outside of China? Can the authors demonstrate that MoRA can perform well in other national geographies?

---

> ### Author Response · Authors · 2025-11-20
> **Response to Reviewer Ay9J - Part 1**
>
> We thank the reviewer for the positive assessment of our work, especially the recognition of our model's strong performance and the rigor of our experimental validation. We are also encouraged that the reviewer appreciates our contribution to the GeoAI community, including our introduction of a high-quality benchmark and, importantly, our being the first to identify scaling laws in geospatial representation learning. We address each of the reviewer's comments in detail below.
> >Q1. **Availability of mobility data, generalizability and reproducibility**:
> >- Heavy reliance on proprietary mobility data: This causes concerns related to reproducibility. I don't believe this resource is publicly available
> >- Single country mobility data: The mobility data is unique to china, casting fairly large doubts on the generalizability of the method. For example, does this method generalize to Europe or the US?  Can the authors demonstrate that MoRA can perform well in other national geographies?
>
>
> **A1:** We thank the reviewer for raising these questions.
>
> First, anonymized mobility datasets, after being researched for about 20 years, exist in many countries. For instance, entities like Google [1], SafeGraph [2], and telecom providers [3] provide high-quality, anonymized mobility datasets in both developed and developing countries for research purposes. Second, while the current work uses online-payment-based mobility data, our framework is adaptable to various alternative data sources, such as vehicle trajectory data, social media check-in data, mobile signaling data, or online search datasets. Therefore, conceptually, the method can generalize to other geographies with any kind of mobility data available locally. International validation is a future step in planning.
>
> We have made every effort to ensure maximum reproducibility despite the proprietary nature of the training data. We have **open-sourced the complete codebase, downstream evaluation datasets, and a high-performance, distilled version of our model**. Crucially, this distilled model allows researchers and end-users to generate geographic embeddings **directly from arbitrary coordinate inputs, completely bypassing the need for the proprietary training data** for both verification and application. We believe these open-sourcing efforts establish a high standard for practical deployment and reproducibility in the field.
>
> >Q2:  What about rural areas where human activity is sparse?
>
> **A2:** We appreciate this important question regarding performance in **rural or sparsely populated areas**. In our framework, the **mobility network** serves as a critical proxy for **long-range geographic connections**, allowing the model to fuse cross-modal signals in a way that transcends mere local physical geography. While areas with extremely sparse human activity lack this specific mobility information, our method remains robust in non-urban settings because the Graph Neural Network (GNN) effectively propagates and captures the underlying spatial semantics through even these sparse, established connections.
>
> While explicitly dichotomizing the entire study area into binary "urban" vs. "rural" categories is non-trivial, we leveraged the vast spatial coverage of our dataset to address this concern. We evaluated the model across provinces with distinct developmental stages to proxy areas of varying human activity intensities. We selected three provinces with contrasting characteristics:
> - Jiangsu & Guangdong: Highly urbanized provinces characterized by dense mobility patterns and developed infrastructure.
> - Sichuan: A province representing lower development intensity, sparser mobility patterns, and varied topography (including mountains and plains).
>
> As quantified in Table I, these regions exhibit significant diversity in data density (check-in counts, population size, and POI density) and spatial distribution (as indicated by Moran's I index [4]). Despite this heterogeneity, Table II shows that MoRA maintains remarkably consistent performance across all three provinces (performance gap $\Delta$< 5% relative to the national benchmark). These results strongly demonstrate our method's effectiveness and robustness even in regions with varying and often sparse degrees of human activity.
>
> Table Ⅰ. Regional descriptive statistics.
> ||**Population density (people/km²)**|**Checkin counts Mean**|**POI counts Mean**|**Population density Moran's I**|**Checkin counts Moran's I**|**POI counts Moran's I**|
> |-|-|-|-|-|-|-|
> |Sichuan|163|521,749|712|0.395|0.410|0.346|
> |Guangdong|493|2,202,197|3,076|0.458|0.512|0.448|
> |Jiangsu|616|1,761,805|2,693|0.231|0.332|0.239|
>
> Table Ⅱ. Regional validation results (R²).
> ||**POP**|**EDU**|**ELD**|**HSR**|**CRI**|**NTL**|**HOU**|**COS**|**Avg.**|
> |-|-|-|-|-|-|-|-|-|-|
> |Sichuan|0.716|0.751|0.881|0.755|0.765|0.465|0.467|0.865|0.708
> |Guangdong|0.847|0.772|0.740|0.800|0.824|0.707|0.599|0.888|0.772
> |Jiangsu|0.758|0.818|0.748|0.860|0.749|0.617|0.581|0.891|0.753

---

> > ### Author Response · Authors · 2025-11-20
> > **Response to Reviewer Ay9J - Part 2**
> >
> > >Q3: The new benchmark primarily describes human-centric socio-economic tasks. It would make sense that mobility data would be more useful here (as demonstrated in their experiments). Other tasks such as land cover classification would likely see better results under the more Earth-observation-centric representation learning strategy. Unfortunately, this trade-off is left unexplored.
> >
> > **A3:**
> > We appreciate the reviewer’s **insightful observation**. We fully agree that human-centric and Earth-observation-centric paradigms offer distinct advantages depending on the application scenario, and that exploring this trade-off would further strengthen and comprehend the current work.
> >
> > To investigate the trade-off, we performed an extended experiment on land cover classification, benchmarking MoRA against several leading models. We utilized the MODIS Land Cover Type product (MCD12Q1, Collection 6.1), an annual global land-cover map at 500 m spatial resolution. Using Land Cover Type 1 (17 classes), we uniformly sampled points within the study area and trained a downstream Multi-Layer Perceptron (MLP) to predict the land-cover classes.
> >
> > Table III shows that MoRA achieves the second-highest performance among the four state-of-the-art methods evaluated. Crucially, it is important to contextualize this result: The top-performing method, AlphaEarth, aggregates nearly all common remote sensing imagery, covering a vast array of spectral bands. In sharp contrast, MoRA achieves a comparable performance relying only on 3-band visible light data (RGB).
> >
> > Such results demonstrate MoRA’s inherent robustness and efficiency, even on tasks outside its primary socio-economic domain. Since the core contribution of this paper is the **cross-modal fusion framework itself**, we plan to incorporate richer spectral data in future work to further bridge the performance gap between more Earth-observation-centric representations and more human-centric representations.
> >
> > Table Ⅲ. Comparison on land cover classification tasks (Acc.).
> > |  | **Dim** | **Land Cover Classification**  |
> > | ------ | ------ |------ |
> > | AlphaEarth|64|0.73|
> > | GeoCLIP | 512|0.67 |
> > | SatCLIP | 256|0.68 |
> > | **MoRA** | 128|0.69 |
> >
> > ---
> > [1] Bassolas, Aleix, et al. "Hierarchical organization of urban mobility and its connection with city livability." Nature communications. 2019. \
> > [2] Nilforoshan, Hamed, et al. "Human mobility networks reveal increased segregation in large cities." Nature. 2023. \
> > [3] Chang, Serina, et al. "Mobility network models of COVID-19 explain inequities and inform reopening." Nature. 2021. \
> > [4] Moran, P. A. P. (1950). Notes on continuous stochastic phenomena.

---

### Official Review · Reviewer_5Ex7 · 2025-11-01

**Soundness:** 3
**Presentation:** 3
**Contribution:** 3
**Rating:** 8
**Confidence:** 4

**Summary:**

The paper proposes a graph-based location representation learning framework that is based on relative location transitions from one (hexagonal) grid cell to another. The framework, despite architecturally complex, is well justified and experimentally strong across spcial and economic tasks. While I would also appreciate benchmarking on natural downstream tasks (mentioned in Figure 1, but not benchmarked against), I understand that the premise of mobility mainly works on social and economical processes.

Overall, I find the paper well-written and well-presented and the results convincing.

**Strengths:**

* Clear and well-justified framework for graph-based location representation learning via relative transitions across hexagonal grid cells.
* Strong empirical performance on spatial and socioeconomic downstream tasks.
* Well-written and clearly presented.

**Weaknesses:**

* Fairly complex methodology including single-modality encoders and a separate graph neural network. A joint unifying architecture would be less engineering-focused. However, ablations justify the individual components well
* It would have been nice to also capturing natural tasks, where mobility matters. For instance, species distribution modelling, like iNaturalist of BirdSnap would be good choices here

**Questions:**

* With MORA, would it be possible to pre-compute and store embeddings for all H3 cells in a region — similar to how AlphaEarth maintains a precomputed embedding database?
* At inference time, does one need to supply POI/satellite imagery or demographics to obtain an embedding, or can the trained GNN alone generate embeddings without requiring additional data downloads?

---

> ### Author Response · Authors · 2025-11-20
> **Response to Reviewer 5Ex7**
>
> We are VERY grateful for the reviewer's positive evaluation. We find the comments highly constructive and insightful. Below, we address the remaining comments one by one:
>
> >Q1. Fairly complex methodology including single-modality encoders and a separate graph neural network. A joint unifying architecture would be less engineering-focused. However, ablations justify the individual components well
>
>
>
> **A1:** We thank the reviewer for this constructive comment. While we agree that a unified architecture could appear less engineering-focused, our current modular framework is deliberately designed for scalability and flexibility. By employing independent encoders, we mitigate potential convergence issues that often arise in highly coupled, multi-modal systems. The current design can be **readily extended** to incorporate new or different modalities in future work without requiring a major architectural overhaul.
>
> >Q2. It would have been nice to also capturing natural tasks, where mobility matters. For instance, species distribution modelling, like iNaturalist of BirdSnap would be good choices here
>
> **A2:**  We thank the reviewer for this insightful suggestion. We have carefully investigated both the iNaturalist and BirdSnap datasets but found that the samples mostly concentrate in North America and Europe. In our current study area, China, the sample size is insufficient to train a classification model of more than 500 categories. Although the current work primarily focuses on urban areas where human activities concentrate, we fully recognize the importance of natural tasks and will continue to seek suitable datasets.
>
> >Q3: **Inference and practical usage**: With MORA, would it be possible to pre-compute and store embeddings for all H3 cells in a region — similar to how AlphaEarth maintains a precomputed embedding database? At inference time, does one need to supply POI/satellite imagery or demographics to obtain an embedding, or can the trained GNN alone generate embeddings without requiring additional data downloads?
>
> **A3:**  Yes, MoRA is designed to support the **pre-computation and storage of embeddings** for all H3 cells in a region, similar to AlphaEarth.
> To enable data-free inference, we have developed a high-performance, knowledge-distilled version of the model that enables direct embedding queries using only geographic coordinates (Latitude/Longitude). This means that at inference time, the model does not require any raw data inputs such as POI, satellite imagery, or demographics—the trained model can operate independently. Thanks to your question, we have added a subsection (Lines 388-394) to better explain the entire procedure from pre-computation to inference.

---

### Author Response · Authors · 2025-12-02
**Rebuttal Summary**

We sincerely thank all the reviewers for their thoughtful comments and suggestions on our work.

We are pleased that reviewers view our work positively, noting its novel and well-motivated framing *(Reviewer adgb)*, rigorous and comprehensive experiments *(Reviewers 5Ex7, Ay9J, adgb)*, convincing empirical results *(Reviewers 5Ex7, Ay9J, L9Si)*, and significant contribution to the GeoAI community *(Reviewers Ay9J, adgb, L9Si)*.

We have carefully addressed each of the reviewers' remaining concerns. To assist the Area Chair in your final assessment, we compiled a summary of the key questions below:


| Reviewers | Comment | Response |
| ------ | ------ | ------ |
| adgb | Required replacing the **mobility graph** with a **neighbor graph** to further validate model effectiveness. | We verified robustness by replacing the mobility graph with a neighborhood graph; **our method maintained superior performance**.|
| Ay9J, L9Si | Requested proof of efficacy across diverse human activity and geographical settings. | We provided a detailed response, offering both **theoretical justification** (on MoRA's dataset-agnostic and prior-free design) and **empirical evidence** (new experiments confirming robustness) across varied geographies and activity levels. |
| 5Ex7, L9Si | Questioned the necessity of **POI/image embedding** and **mobility network** data at inference/deployment. | We developed and open-sourced a knowledge-distilled version of the pre-trained MoRA that supports **direct inference using only geographic coordinates** (latitude and longitude), with no additional inputs. We added a new subsection for clarity in the main text (Ln 388-394). |
| Ay9J | Requested discussion of whether strong human-centric performance comes at the expense of other tasks (e.g., land cover classification), where Earth‑observation‑centric representations may excel. | We **benchmarked** MoRA against leading Earth-observation-centric models on a land cover classification task, demonstrating its robustness across downstream domains. |
| L9Si  |Questioned our use of LightGBM as the **downstream evaluator** instead of linear models. | We clarified that using linear models is **not a universal practice**, even in the references cited by the reviewer. We justified the use of LightGBM to ensure a **fair comparison of optimal embedding performance**. |


Again, we are very grateful for the reviewers' comments, which we believe further strengthen the robustness of this work and the clarity of the manuscript.

We thank the Area Chair for the time and care devoted to overseeing the process and considering our submission.

---

### Meta-Review · Area_Chair_BMr3 · 2026-01-07

**Summary:**

Reviewers broadly agreed that the paper addresses an important and timely problem in region representation learning and presents a technically solid and well-executed approach. Key concerns regarding reliance on mobility graphs, inference-time practicality, and task specialization were addressed. Some concerns remain partially unresolved, particularly regarding cross-country generalization beyond China, reliance on proprietary mobility data, evaluation protocol choices (e.g., LightGBM vs. linear probing), and the degree of conceptual novelty relative to prior multimodal region-embedding work. Given that the main concerns were addressed and considering the overall contribution, I recommend acceptance.

**Reviewer Concerns:**

Resolved concerns:

1.Reliance on mobility graphb versus simpler neighborhood graphs

Reviewer adgb questioned whether MoRA’s effectiveness critically depends on proprietary mobility graphs and requested an ablation replacing the mobility graph with a simple neighborhood adjacency graph. The authors added the requested ablation and showed that while performance degrades when replacing the mobility graph, MoRA still performs competitively, and the full mobility graph consistently yields superior results. The additional experiment directly addresses the concern and clarifies both the value of mobility edges and the viability of a neighborhood-graph fallback.

2.Necessity of multimodal inputs and mobility data at inference time

Reviewers 5Ex7 and L9Si questioned whether MoRA requires POI, imagery, demographic data, or mobility networks during inference, which would limit practicality. The authors clarified that inference does not require these inputs and released a knowledge-distilled version that maps geographic coordinates directly to embeddings. They added a clarifying subsection to the paper. The distilled coordinate-only inference model directly addresses deployment and practicality concerns.

3.Tradeoff between human-centric tasks and Earth-observation-centric tasks

Reviewer Ay9J asked whether MoRA’s strong performance on human-centric socio-economic tasks comes at the expense of performance on Earth-observation-oriented tasks such as land cover classification. The authors added a benchmark on land cover classification against EO-centric models, showing that MoRA is competitive, though not the top performer. The additional experiment demonstrates that MoRA is not narrowly specialized, though EO-centric models with richer spectral inputs still retain advantages for purely physical tasks.

Unresolved or partially resolved concerns:

1.Data Issue, Efficacy across diverse human activity levels and geographies

Reviewers Ay9J and L9Si questioned whether MoRA generalizes across regions with varying human activity density and across different geographic settings. Reviewer Ay9J explicitly raised concerns about reliance on proprietary WeChat Pay mobility data and reproducibility. Reviewer L9Si also framed this as diminishing generalization and practicability. The authors provided theoretical arguments about the dataset-agnostic nature of the framework and added new experiments across provinces with substantially different development levels and mobility sparsity.

The distilled coordinate-only model helps use embeddings, but it does not allow independent retraining or verification of the full pipeline. And the claim that mobility data exists elsewhere does not guarantee similar graph structure, coverage, or noise properties.  The new experiments convincingly demonstrate robustness across diverse regions within China. The rebuttal argues conceptual portability and the availability of mobility data elsewhere, but does not empirically validate cross-country generalization.


2. Choice of LightGBM instead of linear probing for downstream evaluation

Reviewer L9Si questioned the use of LightGBM rather than linear models for downstream evaluation. The authors argued that linear probing is not a universal standard and justified LightGBM as a consistent and fair way to evaluate the optimal predictive capacity of embeddings across all baselines. The justification is reasonable and consistently applied, partially resolving the concern, but some readers may still prefer a inear-probe comparisons for convention and interpretability.

3. Novelty and technical contribution skepticism (Reviewer L9Si)

Reviewer L9Si questioned whether MoRA’s technical contribution is fundamentally new, arguing that combining mobility, POI, and imagery is common in region representation learning and that MoRA mainly scales this paradigm. The author defended novelty via conceptual framing (mobility as backbone, spatial tokens, scaling laws), but not convincing enough to resolve the concern.

**Reviewer Scores:**

Reviewer 5Ex7 would likely remain the score of 8, because all the concerns are resolved.

Reviewer adgb would most likely remain the score of 6, because the most important concern, the lack of a neighborhood graph ablation, is effectively resolved.

Reviewer Ay9J would likely to maintain the score of 6. Reviewer Ay9J raised deep concerns about reproducibility, reliance on proprietary mobility data, and cross-country generalization, and some concerns are not resolved. For example, while the authors added within-China regional experiments and argued dataset-agnostic portability, they did not empirically validate cross-country generalization.

Reviewer L9Si will likely maintain the score of 4. Reviewer L9Si expressed concerns about the technical novelty, evaluation protocol, country-limited scope, and inference clarity. While the authors clarified inference and defended the use of LightGBM, they did not provide linear-probe comparisons, did not empirically demonstrate cross-country generalization, and did not convincingly address the reviewer’s claim that the contribution is largely an incremental scaling of existing multimodal region-embedding paradigms.

---

### Decision · Program_Chairs · 2026-01-26

Accept (Poster)